# The Known and Unknown “Knowns” of Human Susceptibility to Coccidioidomycosis

**DOI:** 10.3390/jof10040256

**Published:** 2024-03-28

**Authors:** Amy P. Hsu

**Affiliations:** Laboratory of Clinical Immunology and Microbiology, National Institute of Allergy and Infectious Diseases, National Institutes of Health, Bethesda, MD 20892, USA; amy.hsu@nih.gov

**Keywords:** *Coccidioides*, coccidioidomycosis, fungal susceptibility, genetic variation, risk factor

## Abstract

Coccidioidomycosis occurs after inhalation of airborne spores of the endemic, dimorphic fungus, *Coccidioides*. While the majority of individuals resolve the infection without coming to medical attention, the fungus is a major cause of community-acquired pneumonia in the endemic region, and chronic pulmonary and extrapulmonary disease poses significant personal and economic burdens. This review explores the literature surrounding human susceptibility to coccidioidomycosis, including chronic pulmonary and extrapulmonary dissemination. Over the past century of study, themes have emerged surrounding factors impacting human susceptibility to severe disease or dissemination, including immune suppression, genetic susceptibility, sex, pregnancy, and genetic ancestry. Early studies were observational, frequently with small numbers of cases; several of these early studies are highly cited in review papers, becoming part of the coccidioidomycosis “canon”. Specific genetic variants, sex, and immune suppression by TNF inhibitors have been validated in later cohort studies, confirming the original hypotheses. By contrast, some risk factors, such as ABO blood group, Filipino ancestry, or lack of erythema nodosum among black individuals, are repeated in the literature despite the lack of supporting studies or biologic plausibility. Using examination of historical reports coupled with recent cohort and epidemiology studies, evidence for commonly reported risk factors is discussed.

## 1. Background

*Coccidioides* spp. are dimorphic fungi endemic to the deserts of the Southwestern United States, Mexico, and Central and South America. In the soil, the fungus exists as arthroconidia, viable spores resulting from hyphal segmentation. When the soil is disturbed by construction, farming, winds, etc., the arthroconidia may become airborne, allowing inhalation by animals and humans. Due to the small, 2–5 μm size, the arthroconidium can enter deep into the lower airway, settling in terminal bronchioles. If not cleared, the arthroconidium swells, undergoing free nuclear division and developing endospores (Figure 1A). Critically, at this point, the developing spherule is now too large for neutrophil phagocytosis and is approaching the limits of macrophage phagocytosis. Over the course of 2–4 days, the spherule matures, ruptures, and releases hundreds of endospores, each of which can develop into a spherule, establishing an exponential growth of the fungus. The developing primary spherule secretes a metalloproteinase, resulting in immune masking [1], however mature spherule rupture leads to a rapid influx of immune cells (Figure 1B top) predominated by neutrophils (Figure 1B bottom).

Unlike many pneumonia-causing pathogens, there have been no reported cases of human-to-human transmission of *Coccidioides* infections. Disease presentation ranges from asymptomatic in the majority of infections (~2/3 [2,3]), primary pulmonary (self-limiting flu-like symptoms or progressive pneumonia which resolves with antifungal treatment), chronic pulmonary (pneumonia which, despite antifungal therapy, remains present after >12 months of treatment), to extrapulmonary dissemination (DCM) (presence of fungus outside of the lungs, frequently skin, bone, brain, soft tissue) [4]. (Figure 2). Dissemination is estimated to occur in 600–1000 individuals of the ~150,000 believed to become infected each year (0.4–0.6%) [3]. Recently, a new framework was published which divides extrapulmonary dissemination into two categories depending on central nervous system (CNS) dissemination and includes a separate category for death from coccidioidomycosis [5]. This increased granularity in patient description will be beneficial in comparing and grouping cases for additional translational or meta-analysis.

Although 27 states and the District of Columbia consider coccidioidomycosis a reportable disease [6], 97% of reported cases occur in Arizona or California, with roughly 2/3 in Arizona [7] (Figure 3A). As a result, the majority of the economic burden exists in those two states. There has been a documented increased incidence of *Coccidioides* infections in endemic areas both by absolute number of cases (Figure 3B) and incidence by population, from 5.3/100,000 in 1998 to 42.6/100,000 in 2011 [8]. In Arizona, specifically, the rate increased from 84.4/100,000 in 2014 to 144.1/100,000 in 2019 [9]. The economic burden of disease has also increased. Using lifetime costs for 7466 cases diagnosed in California in 2017, Wilson et al. [10] found total lifetime costs in CA to be $700 million ($429 million direct and $271 million indirect). Highest per-person costs were for DCM ($1.025 million), with even uncomplicated coccidioidal pneumonia costing >$22,000 per person. Similar calculations were performed for Arizona [9], in which they calculated the incidence-based lifetime cost for the 10,359 cases in Arizona in 2019 to be $736 million. Not surprisingly, the highest economic burden came from patients with disseminated disease, with direct costs calculated to be $1.26 million (healthcare costs for diagnosis, treatment, procedures and care) and indirect costs to be $137,400 (cost of lost work and mortality) per person [9].

## 2. Exogenous Risks of Infection and Disseminated Disease

In the past two decades, external factors, such as immunocompromised hosts (HIV-AIDS) [11,12] or immunosuppression for medical reasons (transplant recipients [13] and biological response modifiers [14,15]), have been reported in individuals with severe or disseminated disease. Additionally, epidemiologic studies and outbreak reports have supported increased risk associated with occupational exposure associated with agricultural work [16,17], construction [18], and outdoor movie production [18].

The first study examining coccidioidomycosis among HIV-infected patients with a formal AIDS diagnosis identified 7/27 HIV cases with active coccidioidomycosis [19], 6 of whom had a diffuse, bilateral pulmonary infiltrate, 5 of whom had autopsy-proven dissemination, and all of whom died (Table 1). Importantly, in 5/7 patients, the coccidioidomycosis diagnosis preceded or was concurrent with the AIDS diagnosis. This led the authors to suggest the increased frequency was due to disease reactivation or enhanced susceptibility to *Coccidioides* among HIV-infected individuals. Notably, treatment with ketoconazole or amphotericin B in 6 patients led to stabilization or improvement of coccidioidomycosis symptoms. A larger case series from 1990 [11] identified 77 coccidioidomycosis patients who were HIV seropositive. The authors noted that, while 31 patients had a diffuse pulmonary infiltrate (9 with extrapulmonary dissemination) similar to that reported by Bronnimann [19], the remaining patients had a variety of clinical presentations, ranging from isolated positive serology to disseminated disease with meningitis, more similar to immunocompetent coccidioidomycosis patients. Additionally, positive response to antifungals was seen in 45/77 (58%) patients surviving at the completion of the study. In fact, the authors comment that coccidioidomycosis “is not always a relentlessly fatal disease” among HIV-infected individuals [11]. A prospective study [20] of 170 HIV-infected patients (median follow-up 11.3 months; range 0–44) documented active coccidioidomycosis developing in 13, 12 of whom had CD4^+^ counts < 250 cells /µL. Only one patient developed disseminated disease (extrapulmonary lymph node), one patient had positive serology, while 11 had documented pulmonary disease, including 5 with the diffuse pulmonary infiltrate previously described. Kaplan-Meier estimate of *Coccidioides* infection after 41 months was 24.6% (95% CI 8.2–41.1). The authors concluded that coccidioidomycosis is a significant opportunistic infection for HIV-infected individuals within the endemic area [20].

A survey of coccidioidomycosis diagnoses in HIV-infected patients over a 30-month period in Arizona [21] identified 153 patients. The authors reported a 4% incidence of symptomatic coccidioidomycosis among 140 AIDS patients and 0.2% among 13 HIV-infected patients compared with 0.149% in Arizona’s 1995 general population [21]. In 2010, Masannat and Ampel [22] re-examined the HIV/coccidioidomycosis issue using a single-site, case-control study of individuals with HIV infection and active coccidioidomycosis. Importantly, this study was performed after the widespread use of antiretroviral therapy (ART) began. Among 257 HIV patients seen within the 64-month study window, 29 had existing coccidioidomycosis, while 12 patients were newly diagnosed during the study (0.9% annual incidence), much lower than previously reported in 1993 (~25%/41 months) [20] or 2000 (4%) [21] and consistent with that of the general population [22]. Additionally, HIV RNA copies < 50/mL, potent ART, and higher CD4+ counts were significantly associated with less severe disease (positive serology, pulmonary nodule or cavity) compared with those patients with more severe focal or diffuse alveolar disease. The authors concluded that ART therapy has improved clinical presentation and outcomes for HIV-positive patients infected with *Coccidioides*. Together, these studies indicate an increased risk of infection severity among patients who have AIDS, however that risk appears to be normalized with positive response to ART therapy.

Transplant-associated immunosuppression, both hematopoietic stem cell transplantation (HSCT) and solid organ transplantation (SOT), is frequently listed as a risk factor for coccidioidomycosis. The first 3 cases of HSCT and coccidioidomycosis were published in 1993 [23]. Since that time, 8 additional cases have been reported [24,25,26,27,28]. It is important to note that of the 11 cases, 5 were documented to have infection prior to transplant. Three others became ill within 35 days of transplant, despite negative serologies, raising the question of reactivation or new infection. In 2005, Glenn et al. [24] reported 121 consecutive HSCT patients, with only 3 cases of coccidioidomycosis, one fatal DCM, one pulmonary reactivation of existing infection with successful treatment, and one identified pre-HSCT with successful treatment and no reactivation after HSCT. A recent review of endemic mycoses after HSCT concluded these events are uncommon and frequently associated with cessation of prophylaxis or increased immunosuppression for graft versus host disease [29].

The literature on SOT is larger. A 2001 review of case reports and case series on coccidioidomycosis in SOT [13] highlights an increased incidence of coccidioidomycosis present in transplant patients compared with the general population, with the first post-transplant year having higher rates (33/47; 70%) than years 2–5 [13]. Subsequent studies have focused on specific organ transplants. Among 391 liver transplant recipients [30], 15/391 (3.8%) were diagnosed with post-transplant coccidioidomycosis, 10 within the first year. One case had indeterminate serologies prior to transplant while 2 cases had reactivation of infection; 12 were reported to be de novo infections post-transplant. Within the cohort, 25 patients had a prior history or were seropositive and 2 cases developed reactivation of infection (8%). Interestingly, there are 2 recent case series of DCM in end-stage liver disease [31,32], hypothesized to be related to cirrhosis-associated immune dysfunction (CAID), a syndrome of immune dysfunction, immunodeficiency, and systemic inflammation [33]. Whether there is resolution of CAID post-transplant and the timing of such has not been reported suggesting one possible cause of increased coccidioidomycosis after liver transplant. In 2017, Lohrmann [34] reported 174 heart transplant recipients over a 9-year period. Incidence of coccidioidomycosis post-transplant was 0.6% at 1 year and 2.3% at 5 years. A recent report from 2 hospitals in Arizona described 91 SOT recipients developing de novo coccidioidomycosis more than 1 year after transplant [35]. Among the 91 individuals, 51 (56%) had asymptomatic infection, including 8 with pulmonary coccidioidomycosis. Symptomatic pulmonary infection was present in 29 cases (32%) and 5/91 (5.5%) had extrapulmonary dissemination [35]. While the majority of reported cases of coccidioidomycosis occur within the first year of transplant, the authors concluded that the infection remains a cause of morbidity and mortality beyond that time [35].

The first of two large studies of lung transplant recipients [36] reported 189 patients spanning 1985–2009. There were 11 cases of coccidioidomycosis (5.8%) with 10 being pulmonary infections. The authors note that despite the time span of the cases and changing immunosuppressive and prophylactic regimens, antifungal therapy was significantly associated with non-development of coccidioidomycosis. This is supported by the second study by Truong and colleagues [37], in which 493 lung transplant recipients at a single center from 2013–2018 were initiated and maintained on azole prophylaxis. There was a single case of asymptomatic seroconversion (1/493, 0.2%) and one case of pulmonary disease after stopping prophylaxis (1/493, 0.2%), which resolved after resumption of antifungals. Despite the apparent increased risk of infection or reactivation post-transplant, these reports suggest surveillance for pre-transplant *Coccidioides* infection and prophylactic antifungal therapy post-transplant are associated with decreased incidence of severe coccidioidomycosis.

Direct immunosuppression through the use of corticosteroids or biologics targeting specific molecules is frequently noted as a risk for *Coccidioides* infection. The first case series of coccidioidomycosis associated with targeted immunotherapy documented 13 patients using TNF inhibitor (TNFi) treatment for rheumatologic diagnosis [14]. Pneumonia was the predominant presentation, occurring in 9 patients, while disseminated disease was reported in 4 with 2 having fatal infections. Subsequently, retrospective studies have examined presence of coccidioidomycosis among rheumatology patients in Arizona treated with biologic response modifiers (BRMs).

The reported studies have either centered on TNFi [38] or grouped all BRMs together [15,39]. In one review, Blair et al. [40] reviewed reports of coccidioidomycosis with various BRMs. In one study of TNFi recipients within the endemic region [38], 49/1770 (2.8%) had proven or probable coccidioidomycosis, a >10-fold increase compared with reported infections in Arizona during the same time (0.26% (2011) to 0.1% (2017)) [41]. While the increased incidence may in part be due to increased surveillance [42], there were 7 cases of DCM (7/1770, 0.4%), 3 receiving infliximab and 4 receiving adalimumab. There was no difference in incidence of infection between patients receiving etanercept, a soluble TNF receptor, versus other TNF antibodies [38]. There has been one case report of DCM in a patient receiving vedolizumab (α4β7 integrin) for Crohn’s disease [43] and one diagnosed with a pulmonary nodule while receiving abatacept (CTLA4) [39]. In a recent case series of 135 patients receiving ruxolitinib (JAK1/JAK2 inhibitor) [44], 8 developed symptomatic coccidioidomycosis; 4 had coccidioidomycosis preceding ruxolitinib, and each had primary pulmonary disease while on therapy. The remaining 4 developed disseminated disease between 1 month and 1 year from therapy initiation. It is noteworthy that despite mutations in the IL12/IFNγ pathway causing susceptibility to DCM (reviewed in Odio [45]), there have been no reports of patients receiving ustekinumab (IL-12/IL-23 monoclonal antibody) developing coccidioidomycosis; nor have there been reports of patients receiving monoclonal biologics against IL-6 (tocilizumab) or JAK1/JAK3 (tofacitinib).

The most direct evidence for steroid immunosuppression and TNFi therapy impacting *Coccidioides* susceptibility comes from mouse studies. In mice infected with the attenuated *C. posadasii* strain 1038 and allowed to control the infection, introduction of dexamethasone resulted in disruption of granuloma by day 5 and 100% mortality by day 25 [46]. Cessation of dexamethasone treatment after 14 days led to increased survival and decreased fungal burdens, although histopathology data indicated treated mice had not developed new granuloma to control the infection by 6 weeks. A second study examined infection using the attenuated *C. posadasii* strain 1038 in the setting of TNF inhibition. Mice receiving TNFi or dexamethasone beginning 2 days prior to infection had similar 100% mortality rates, with median survival of 37 and 34 days, respectively, compared with 100% survival at day 70 in the control group (Figure 4A) [47]. Stopping TNFi 14 days after infection did not ameliorate the mortality or increased fungal burden seen in mice treated throughout the infection (Figure 4B) [47]. Critically, mice with controlled infections treated with TNFi had significantly increased lung and spleen fungal burdens compared with untreated mice [47]. These studies highlight the role of adequate immune response centered on TNF both at the time of infection as well as ongoing control of existing infections.

## 3. Genetic Variants Associated with *Coccidioides* Dissemination

Identification of causative mutations in patients with invasive fungal disease in isolation or as part of a broader primary immune deficiency has identified key genes, proteins, and pathways critical to fungal resistance. Despite the 600–1000 cases of disseminated infection per year [3], there are only 15 patients in the literature with identified monogenic mutations. The first DCM patient with an identified mutation was a young girl with *Coccidioides* meningitis and hyper-IgE syndrome (HIES) caused by a dominant-negative mutation in *STAT3* [48]. Patients with HIES are susceptible to invasive fungal disease; however, in one series of 64 STAT3-mutated HIES patients, invasive fungal infections only occurred among those with existing lung damage [49]. Since the original patient, two additional HIES patients were reported with *Coccidioides* meningitis [50,51], while we have diagnosed one additional *STAT3*-mutated patient with chronic, refractory pulmonary coccidioidomycosis (Hsu, unpublished). Additional patients reported include cytokine receptor mutations: one with a dominant-negative mutation in *IFNGR1* [52], two siblings with homozygous *IL12RB1* mutations [53], and one patient with heterozygous *IL12RB2* [45] as well as transcription factors: two with gain-of-function *STAT1* mutations [54] and three members of a single family with dominant-negative *STAT4* mutation [55]. One patient has been reported with a dominant *GATA2* mutation [56], which causes the loss of B-cells, NK-cells, monocytes [57], and dendritic cells [58]. One teenage patient presenting with DCM was found to be heterozygous for a truncating mutation in *NFKB2* [59], leading to haploinsufficiency of the non-canonical NFKB signaling pathway [60]. More recently, a young boy with undiagnosed *CTPS1* deficiency presented with mediastinal lymphadenopathy and DCM [61]. It is noteworthy that, with the exceptions of *GATA2, NFKB2*, and *CTPS1*, the mutations occur within components of the innate IL-12/IFNg pathway [Figure 5]. Additionally, despite its seeming importance, mutations within this pathway are exceedingly rare. Little is known regarding the role of NK cells during *Coccidioides* infections.

While the majority of patients reported have been case reports of individuals or families, recently, population variants in *CLEC7A*, encoding the β-glucan pattern recognition receptor DECTIN-1, *PLCG2*, *DUOX1,* and *DUOXA1* were shown to be overrepresented in an Arizona cohort of DCM patients compared with ancestry-based frequencies in gnomAD or 1000 Genomes datasets [4]. These genetic findings were replicated in a larger, validation cohort of DCM patients recruited from southern California. When placing patients with “severe disease”, either disseminated or chronic pulmonary coccidioidomycosis, together and comparing them to the 1000 Genomes dataset, DECTIN-1 variants remained overrepresented, while PLCG2 and DUOX1/DUOXA1 were not, suggesting a role for DECTIN-1 recognition and signaling in control and clearance of the infection, as well as prevention of dissemination. In a similar analysis, which also included patients with primary pulmonary coccidioidomycosis, none of the identified genes or variants were overrepresented, indicating these damaging variants are not infection susceptibility risks but rather control of infection and dissemination. These findings implicate the initial recognition of the ruptured spherule and corresponding immune response as key to controlling the infection. Presence of common, population-level variants associated with susceptibility to disseminated disease is consistent with the geographic isolation of a pathogenic fungus since most individuals will not have the opportunity to be infected.

Innate immune cells recognize pathogens through pattern recognition receptors including Toll-like receptors (TLRs), C-type lectin receptors (CLRs), RIG-I-like receptors (RLRs), and Nucleotide-binding oligomerization domain-like receptors (NLRs). *Coccidioides* cell walls contain β-glucan which is recognized by the CLR, DECTIN-1. Mouse studies have shown that DECTIN-1 is required for resistance to *Coccidioides* [62], and the difference between the susceptible C57BL/6 and the resistant DBA/2 strains is associated with a splice variant in the C57BL/6 strain which deletes the stalk region, placing the β-glucan-recognizing C-type lectin domain closer to the cell surface [63]. TNF production in response to formalin-killed spherules was significantly reduced by the shorter splice form [63]. Fungal recognition of β-glucan by DECTIN-1 activates Src kinase, phosphorylating DECTIN-1 and activating spleen tyrosine kinase (SYK) [64]. Active SYK recruits the CARD-9/BCL10/MALT1 complex (CBL) activating NF-κB. Concurrently, SYK phosphorylates PLCG2 leading to increased intracellular Ca^++^ levels, calcineurin activation, nuclear translocation of NFAT, and NFAT transcriptional activation. Together, NF-κB and NFAT drive transcription of pro-inflammatory cytokines including TNF (Figure 6). Functional studies using peripheral blood mononuclear cells from patients with identified DECTIN-1 and PLCG2 variants demonstrated impaired β-glucan-induced TNF production compared with healthy controls [4].

DUOX1 and DUOXA1 form a heterodimeric NADPH oxidase highly expressed on ciliated epithelial cells in the lung, especially the alveolar epithelial type II cell (AECII) found in the alveolus [65]. Assembly of the heterodimer within the endoplasmic reticulum is required for translocation to the apical plasma membrane. Activation of the complex occurs via two Ca^++^ sensing EF-hand domains in the first intracellular loop of DUOX1, after which electron transport through the complex releases hydrogen peroxide (H_2_O_2_) into the luminal space. Importantly, DECTIN-1 is also expressed on the apical surface of pulmonary epithelial cells and engagement of DECTIN-1 with depleted zymosan, a DECTIN-1 specific agonist, is sufficient to induce H_2_O_2_ production by DUOX1/DUOXA1 (Figure 7A) (Adapted from Hsu et al. [4]).

Considering the initial infection is pulmonary, it is plausible that most people exposed are not infected due to the arthroconidia being swallowed or ending in a larger airway where it would be cleared by ciliary motion. By contrast, arthroconidia inhaled into an alveolus are able to undergo spherule maturation. Upon spherule rupture, fungal recognition by DECTIN-1 on alveolar macrophages leads to NF-kB activation and TNF production, simultaneously activating and releasing H_2_O_2_ to the alveolar space. While it is known that DUOX1-derived H_2_O_2_ interacts with lactoperoxidase and thiocyanite to form antimicrobial hypothiocyanite [66], multiple cellular models demonstrate that H_2_O_2_ itself activates NK-kB in a dose- and time-dependent manner, leading to p65 nuclear translocation [67]. Co-treatment of epithelial MCF-7 cells with H_2_O_2_ and TNF increased NFkB response 3-fold over H_2_O_2_ or TNF alone [68]. These studies, coupled with the genetic findings of *CLEC7A*/DECTIN-1, *PLCG2*, *DUOX1*, and *DUOXA1* variants in DCM, highlight the role of the pulmonary epithelia in immune surveillance, initial response, and AECII/alveolar macrophage crosstalk mediated by H_2_O_2_ (Figure 7B).

## 4. Centrality of TNF

In the severe and disseminated cases from genetic studies noted above [4], breakdown of the DECTIN-1/PLCG2 pathway led to decreased TNF production by peripheral blood mononuclear cells. Mouse studies revealed that blockade of TNF signaling inhibits initial response to infection, leading to dissemination and mortality [47], and impairs control of existing infections, causing breakdown of granuloma and increased fungal burden [47]. Cohort studies involving biologic therapies revealed that inhibition of TNF by monoclonal antibodies or soluble TNF receptors led to an increased risk of coccidioidomycosis [14,15]. TNFi-induced reactivation of controlled disease was originally reported for tuberculosis (TB) [69]. The relative risk of TB infections after TNFi, either new or reactivation, is increased up to 25 times [70], leading the European TBNET consortium to suggest screening of all adult candidates for the presence of latent TB infections prior to TNFi initiation [70]. Similar prospective studies of patients within the *Coccidioides* endemic region would be beneficial in firmly defining the risk and extent of disease for both *Coccidioides* naïve and previously infected individuals.

## 5. Additional Risk Factors

Within the body of literature surrounding coccidioidomycosis, increased risk of infection is reported in distinct populations including male sex [2,16,71], pregnancy [72], individuals of African [2,16,73,74] or Filipino [75,76,77] descent, and type B blood group [78,79,80]. Many of these studies are from the original ground-breaking epidemiologic work performed by Myrnie Gifford, Charles Smith, Demosthenes Pappagianis, Paul Williams, Rodney Beard, Margaret Saito, and others. These studies provided the first glimpses into the types of infections, populations infected, and infection severity. They assembled cohorts of tens to a few hundred individuals, manually curated hospital records, and reported the findings. Currently, these studies, or their conclusions, remain the citations used in many review articles and introductions to research articles. Below, each of these claims will be reviewed in light of recent studies.

## 6. Male Sex

From some of the earliest studies, the male predominance among cohorts was noted. In fact, in 1914, MacNeal and Taylor [81] stated, “In the first place it is a disease of males, and almost exclusively adult males”. They noted that of 31 reported cases, there were only one female and one child. Mid-20th century studies using military bases in the endemic area additionally reported male predominance [2]. Durry et al. [82] repeated these observations in a 1991 study of 128 individuals with acute coccidioidomycosis by reporting a relative risk of 2.5 (95% CI 1.2–5.0) for male sex. Similarly, Bays et al. [83] used the Veterans Affairs–Armed Forces Database to retrospectively study disease incidence in the mid-1950s. Of 531 coccidioidomycosis patients, there were 69 cases of DCM, all of whom were male. More recently, large-scale epidemiologic studies from California [16] and a review of published large cohorts [71] have confirmed a relative risk of infection among men. A meta-analysis demonstrated similar rates of infection between young men and women; however these diverge, with women’s risk declining beginning in early to mid-teens, with significant differences by age 19 [71]. Increased rates of infection were also observed in a colony of non-human primates, in which 22/40 males versus 1/17 females were diagnosed with coccidioidomycosis [71]. Similarly, male dogs had increased infections reported compared with females [71], although this risk was ameliorated in castrated males who had risk levels similar to female dogs [71]. Given the onset of risk in mid to late teens, corresponding to onset of puberty, and protective effect of castration in males, increased sex hormones, such as testosterone, may be relevant.

## 7. Pregnancy

Pregnancy is often listed as a risk factor for disease severity. Resolved infections prior to pregnancy do not appear to provide an increased risk for reactivation [84] however initial infection during pregnancy appears to provide increased risk. The most recent review of the literature documented 4/8 (50%) first trimester cases, 8/13 (62%) second trimester cases, and 24/25 (96%) third trimester cases presenting with disseminated disease [72]. It has been suggested [84,85] that the increased risk during later gestation and post-partum is related to an immune reconstitution syndrome. Given the protective effect castration had on male dogs developing disseminated disease [71], it is possible the increased levels of testosterone during pregnancy may contribute to the increased risk of disease. Testosterone increases during pregnancy, with total testosterone increasing during the first trimester and remaining elevated, while free testosterone is 2–3 fold higher during the third trimester [86,87]. Supporting a potential role for testosterone, one case report of peritoneal coccidioidomycosis occurred following fertility treatment with clomiphene citrate [88], which is documented to increase testosterone levels and, in fact, is used off-label to treat hypogonadism and low testosterone in men [89].

## 8. African American Susceptibility

Numerous studies have documented differences in disease severity among African American individuals. Early cohort studies frequently focused on military personnel in Southern California, allowing for the prospective studies of (frequently *Coccidioides* naïve) individuals receiving similar diet, housing, and access to medical care. Despite similar rates of coccidioidin skin-test conversion to positivity as a marker for infection over a 3-month span, 4/49 (8%) of African Americans presented with disseminated disease compared with 0/34 Caucasians [90]. After a 1977 dust storm near Bakersfield, CA, there was a disproportionate amount of dissemination among African Americans (25–50% [75,76,91]) and Asians (38% [76]) compared with 0% dissemination among Caucasian patients [76].

Similar burden of severe disease has been seen in epidemiologic studies focusing on hospitalization. In one study, African Americans had a 12-fold higher rate of hospitalization for disseminated coccidioidomycosis in Arizona than Caucasians despite equivalent all-cause hospitalizations [74]. Those findings were matched in California, with 8.8-fold higher hospitalization for DCM for African Americans compared with Caucasians [74]. A later study evaluated data for hospitalizations in Arizona and California from 2005–2011 and compared the annual coccidioidomycosis hospital incidence rate per 100,000 persons [92]. While the rates fluctuated year to year, Asian/Pacific Islander (11.8/100,000 persons, 95% CI 9.1–14.4) and African American (12.8/100,000 persons 95% CI 10.9–14.6) individuals had consistently higher incidence of hospitalization for coccidioidomycosis than individuals of other ancestries (Caucasians, 6.2/100,000 persons (95% CI 5.2–7.2); Native American, 5.6/100,000 persons (95% CI 4.2–7.1); Hispanics, 5.2/100,000 persons (95% CI 4.3–6.1); and 0.5/100,000 persons (95% CI 4.1–7.1) among other race/ethnicities) [92] (Figure 8A). More recently, a large cohort of coccidioidomycosis patients from southern California was reported, which again highlighted the difference in disease presentation across individuals of different genetic ancestries (Figure 8B) [4]. While these studies do not adjust for socioeconomic factors, there was no significant difference across quartiles of household incomes in hospitalization for coccidioidomycosis compared with all-reason hospitalization [92], suggesting some inherent basis for susceptibility to dissemination or severe disease.

## 9. Filipino Susceptibility

Increased risk of disease among individuals of Filipino ancestry has been frequently cited in the literature. This stems from a report of coccidioidomycosis in Kern County by Gifford in 1936 [93], in which the rate of disseminated cases per 100,000 of the at-risk population for various races was calculated. She found that Mexican-Americans were 3 times, African Americans 14 times, and Filipinos 175 times as likely to have disseminated disease as whites. Notably, during that time, there were large numbers of Filipino farm workers in the Central Valley of California, allowing the possibility of occupational exposure rather than inherent genetic risk factors to be responsible. Three studies conducted after the 1977 California dust storm reported 225 individuals at Lemoore Naval Air Station [94], 230 individuals residing in the non-endemic area affected by the dust storm [94], and 18 individuals at Lemoore diagnosed 2–4 weeks following the dust storm [75]. Across these 3 studies, 4/19, 2/3, and 1/4 Filipino individuals exhibited extrapulmonary dissemination. Based on these 4 isolated studies with small sample sizes, the idea of increased DCM incidence among Filipinos has been repeated in the literature. Since that time, higher risk of dissemination among Filipinos has often been cited, without rigorous epidemiologic study.

## 10. Erythema Nodosum and *Coccidioides*

Differences in skin manifestations of *Coccidioides* infections among African Americans have been cited. The presence of erythema nodosum (EN) during coccidioidomycosis was reported in 52/1218 white coccidioidomycosis patients compared with 0/92 African American individuals [2]. In a retrospective chart review, 23/43 Hispanic and 5/8 Caucasian pregnant women were noted to have EN compared with 1/9 African American women [95]. Within that study, no patient with EN developed disseminated disease, leading to the conclusion that EN is a marker for positive disease outcomes for pregnant women [95]. The absence of EN has been speculated to be related to increased rates of dissemination and disease severity among African Americans [96]. This conclusion is confounded however by two facts. First, dermatologic findings in patients with darker skin are frequently missed [97]; in a retrospective chart review, such as the pregnancy study [95], absence of chart notation is not the same as absence of the symptom, and yet it is inferred that EN was not present. Second, we now know that there is increasing disease severity over the course of pregnancy, with later infections having more severe outcomes; unfortunately, the study does not report the trimester with presence of EN.

An additional possibility is highlighted in three cases of EN presenting prior to onset of seropositivity or pulmonary symptoms [98]. EN is believed to be a type IV delayed hypersensitivity response to soluble antigens and requires sensitized T lymphocytes. This raises the possibility that EN occurs in the setting of an individual previously infected, such as those individuals who are coccidioidin skin test positive, despite never having been symptomatic. Blair et al. [99] reported serological responses in immunocompetent patients (*n* = 298) for 12 months after symptom onset. For each serologic assay—IgG serology, complement fixation, and immunodiffusion, only ~40% of individuals maintained seropositivity by one of those tests, and ~70% were seropositive by any of the tests. This demonstrates a decline in seropositivity after infection despite the presence of CD4+ T cells which respond to the coccidioidal antigen, T27K [100]. Additionally, skin test data indicates 80% of individuals residing in the endemic region for 5 years or more will have a positive coccidioidin skin test [101]. If the presence of EN is a marker of previous infection, then it should be associated with less severe disease due to an already present adaptive immune response. In one original report from the 1977 California dust storm, 6/18 cases had EN or maculopapular eruption, none of whom disseminated [75]. Notably, in addition to the pregnancy study cited above [95], an outbreak in rural New Mexico occurred, in which 25 individuals developed EN with no identifiable cause [102]. Among those EN cases, 17/20 reported being present at the same festival construction site, 16/20 noted joint pain and fatigue, and 9/15 had abnormal chest radiographs, including nodules and consolidation suggestive of *Coccidioides* infection [102]. Within one urgent care system, 170/176 patients presenting with EN did not have an associated pneumonia diagnostic code; 44 of them (44/176, 25%) had coccidioidal serology testing, with 27/44 (61.4%) returning positive tests [103]. If EN is, in fact, the result of previous exposure to *Coccidioides*, then one possible study would be to assess the T-cell response in patients who present with EN in the endemic region. Prior exposure should result in IFN-γ release in response to *Coccidioides* antigen.

## 11. Blood Group Association

A frequently cited risk association has been type B blood group. One of the original studies describing this [78] performed blood group typing on 105 patients, 57 with pulmonary disease and 48 with DCM. Comparing numbers of patients in each blood group with DCM vs. pulmonary, the authors stated that individuals with type B are twice as likely to have dissemination (4 pulmonary vs 8 DCM). A re-analysis of the data presented indicates that dissemination in type B (8/12) is not statistically different from patients with type O (24/54, *p* = 0.2095, Fisher’s exact) or the remaining non-B blood types (40/93, *p* = 0.1374, Fisher’s exact). Further, the authors note the rate of type B blood is 2-fold higher in African Americans and 2.7-fold higher among Filipinos compared with whites in California. The next study was 18/260 renal transplant patients [79]. Among the 13 patients with DCM, 3 had type B and 2 had type AB blood. The authors combine these to state that 41.7% [sic] of patients had blood group B antigens, statistically higher than the 13.5% of transplant recipients [79]. A third study regarding type B risk reported 6 Hispanic DCM patients vs 0 pulmonary patients with type B and claimed increased risk [80] although no increased risk was observed among white or African Americans with type B. Despite small studies and flawed interpretation of the data, the myth of increased susceptibility of type B blood group to dissemination is still repeated in review articles.

## 12. Unanswered Questions

This review has focused on human susceptibility to *Coccidioides* and highlighted the various ways immune recognition and surveillance may be impaired through extrinsic factors, genetics, and additional host factors. On the host side, cells involved in response to the initial infection are not fully characterized. While alveolar macrophages are present and neutrophils arrive rapidly upon spherule rupture, the role of innate lymphoid cells, NK cells, and eosinophils has not been characterized. Petkus and Baum [104] reported that NK cells are the leukocytes responsible for *Coccidioides* growth inhibition in vitro. This conclusion is hampered by their use of the Leu-11 antibody [104], which recognizes CD16 and is found not only on NK cells but also on neutrophils, monocytes, and macrophages. The adaptive T-cell response is critical for establishment of long-term immunity, and optimizing this has been foundational for vaccine development efforts. Additionally, how dissemination occurs and which cells contribute is still unknown. Hampton and Chtanova [105] summarized the numerous immune cells including dendritic cells, neutrophils, monocytes, and macrophages, that can enter the lymphatic network, transit to the draining lymph node, and present antigens. Whether dissemination occurs via this mechanism with spherule rupture in or near the lymph node or by some other, as yet undetermined, method is not known. Utilization of fluorescent *Coccidioides* strains [106] and advanced confocal microscopy techniques may help address this issue. Once the spherule ruptures and phagocytosis of the endospore or spherule occurs, whether or not there is continued maturation of the endospore or spherule intracellularly has not been studied. After infection, as the host response begins containment, how coccidioidal granulomas develop and how sequestration of quiescent spherules is maintained are open fields for investigation. Advances in molecular biology techniques allowing single-cell transcriptomics, spatial transcriptomics, proteomics, and high-resolution microscopy may help answer these questions. Coccidioidal infection and its control are not strictly host-dependent. Molecular studies, including genetic, transcriptional, and proteomic analysis of the fungus, have begun. These were excellently reviewed by Kirkland and colleagues [107], and details will not be covered here. Now that researchers are undertaking these studies however, two important unanswered questions are how the transcriptomic and proteomic findings from cultured organisms compare to parallel studies from infected mouse or patient isolates and how the host response changes during stages of *Coccidioides* development.

African American ancestry has been reported as a risk factor even in stringent epidemiology studies; however, the underlying reason for this has yet to be identified. Whether this is representative of a formerly protective trait that is maladaptive in the setting of Coccidioides exposure (similar to sickle cell trait for malaria), or due to systemic biases in housing, healthcare, or occupation is unclear and should continue to be investigated. To date, there are no studies indicating that recent immigrants or visitors from African countries experience increased rates of dissemination. This suggests a more complicated and nuanced reason for the susceptibility to severe infection seen among African Americans. One possible contributor could be transgenerational transmission of small RNAs and epigenetic modification. This was demonstrated in C. elegans after starvation stress, in which small RNAs targeting genes involved in nutrition were present through three consecutive generations [108]. Additional human studies have documented the effects of the Great Chinese Famine in the late 1950s, altering infection susceptibility among survivors’ offspring, with a 6.5% increase in pulmonary TB among children and an 8.3% increase in grandchildren compared with the expected incidence [109]. Increased susceptibility to sexually transmitted and blood-borne infections was also noted among the grandchildren [109]. With these studies in mind, RNA-Seq after in vitro *Coccidioides* exposure or determination of methylated genomic regions via bisulfite or nanopore-based DNA sequencing could be pursued to address this issue.

## 13. Conclusions

Susceptibility to coccidioidomycosis is complicated—incorporating occupational exposure, duration of time in endemic regions, sex, pregnancy, extrinsic medical conditions discussed here, and genetics. Both novel, pathogenic mutations in innate immune pathways as well as more common, population-level variants impacting fungal recognition have been identified. While there remain significant gaps in our knowledge of *Coccidioides*/host interactions, strong evidence supports the central role of innate immune system functioning. Disruption of fungal recognition or response due to genetic variants, TNF inhibitors or dexamethasone is associated with severe or disseminated disease. Reactivation of controlled infection after TNF inhibitors or immune suppression post solid organ transplant has been reported. Uncontrolled HIV infection with low CD4+ cells and significant viral load are associated with disease, however HIV-positive status itself appears not to increase risk. Epidemiologically, male sex and later stages of pregnancy increase risk of DCM. The extrinsic factors, genetic susceptibility, and additional intrinsic factors discussed within this review are summarized in Figure 9, with the strength of evidence for each indicated.

Researchers should take care when using race/ethnicity as a primary grouping of patients. There is no scientific definition of “Hispanic”, “Filipino”, or “African American”. Most people will agree that “Hispanic” individuals in the endemic area vary widely from “Hispanic” individuals in Florida. Utilizing genetic ancestry rather than self-reported race and ethnicity to group individuals, Filipinos cluster broadly with East Asian individuals [110]. One large study examining > 100,000 individuals in Northern California [111] compared self-reported race/ethnicity to genetic ancestry. Self-identified Filipinos were noted to have more European admixture and a larger spread on principle component analysis than other “East Asian” individuals [111]. By the same token, “African American” includes anyone with ancestry from an entire continent, regardless of how long their ancestors have been in the United States. Given the genetic diversity seen within each population, the assumption of increased risk by “race” needs to be carefully examined. Genetic studies can be designed using case/control matching, in which the most genetically similar controls are selected for each case regardless of self-identified race or ethnicity, as described previously [4].

It may be time to rethink many of the recited risk factors surrounding coccidioidomycosis. Caution needs to be taken when referencing older papers with small study sizes, in which data were extrapolated from tens of cases to determine incidence per 100,000 individuals. The generalizations hypothesized in the papers have now entered the canon of coccidioidomycosis susceptibility, as reviewed above. Smaller, observational studies and case reports are critical for documenting disease, generating hypotheses, and identifying previously unrecognized at-risk patient cohorts. Given the larger studies that can now be performed, findings from small cohorts or historical studies should be replicated with more individuals, utilizing solid statistical methods and incorporating robust underlying science.

## Figures and Tables

**Figure 1 jof-10-00256-f001:**
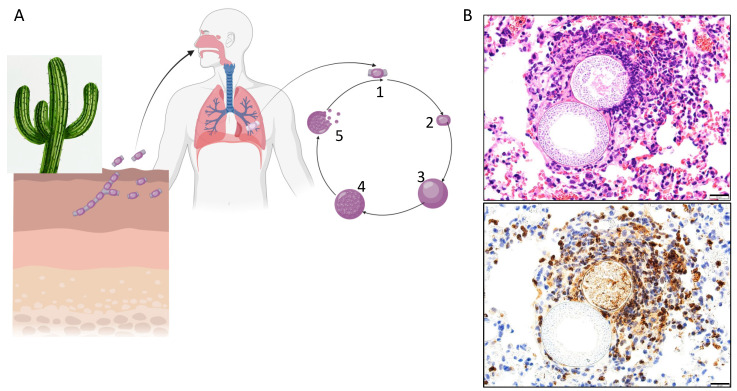
Life cycle of *Coccidioides*. (**A**) In the desert soil of the southwestern US, Mexico, and Central and South America, *Coccidioides* are found as septated hyphae, which break apart into individual arthroconidia. Upon soil disruption, the arthroconidia become airborne and may be inhaled by a mammalian host. Inside the warm, moist environment of the lung, the arthroconidia (1) will swell (2), undergo internal cell division (3), develop endospores (4), then rupture (5), releasing endospores that can develop into new spherules. Figure created using BioRender. (**B**) Mouse lung 14 days post-infection with *C. posadasii*. Top, H&E stain showing immune cells swarming the upper, rupturing spherule (stage 5 on the left), while the lower, unrecognized spherule is less mature (stage 3–4), as evidenced by the open central space. Bottom, MPO staining for neutrophils highlighting immune masking of the unruptured spherule. Size bar = 20 μm.

**Figure 2 jof-10-00256-f002:**
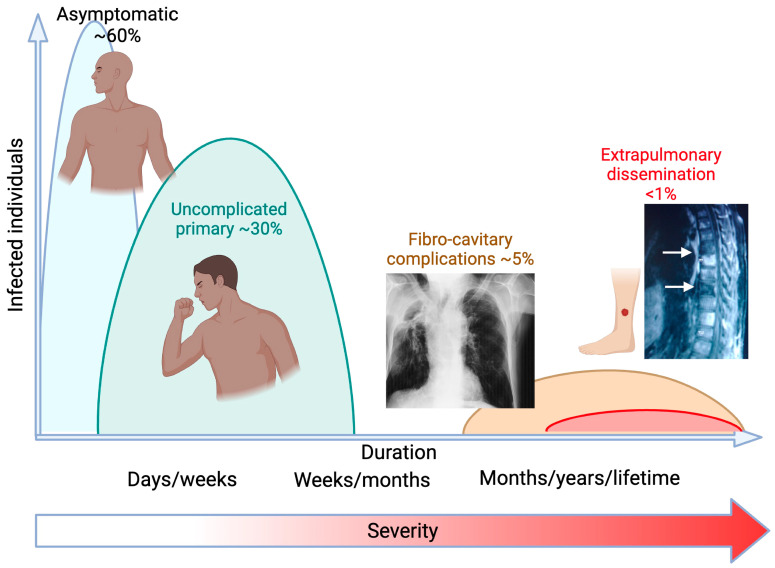
Coccidioidomycosis presentation classifications. The majority of those infected will remain asymptomatic, ~30% will have uncomplicated primary disease. Roughly 5% will present with chronic pulmonary symptoms, including fibrocavitary complications, while <1% will develop extrapulmonary dissemination. Figure created using BioRender.

**Figure 3 jof-10-00256-f003:**
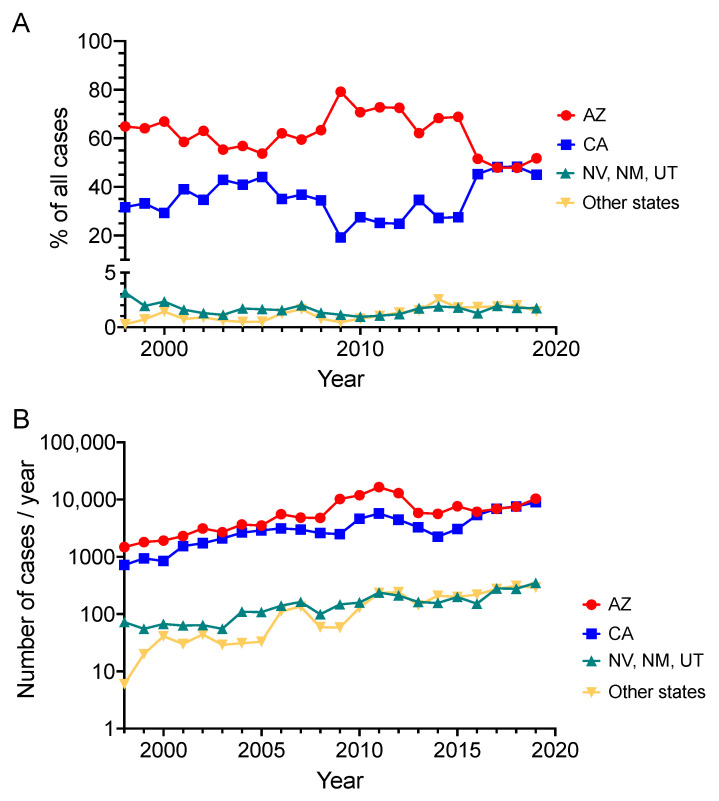
Coccidioidomycosis cases by state. (**A**). Percent of reported coccidioidomycosis cases per year by state, 1998–2019. Data adapted from [6]. (**B**) Annual number of coccidioidomycosis cases reported to CDC, 1998–2019. Data adapted from [7]. AZ—Arizona; CA—California; NV—Nevada; NM—New Mexico; UT—Utah.

**Figure 4 jof-10-00256-f004:**
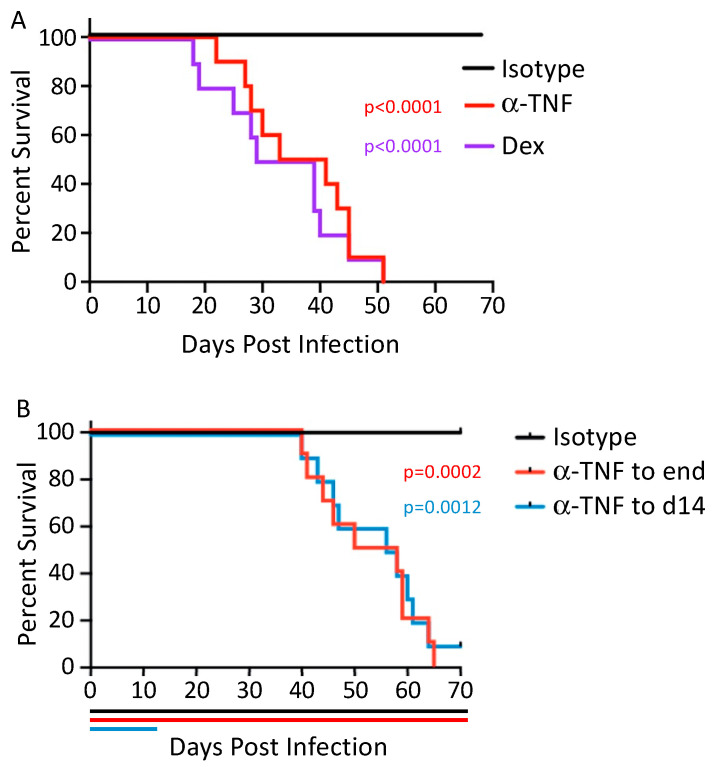
Impact of TNF inhibition or steroids on establishment and maintenance of control of *Coccidioides*. (**A**) Mice were treated with anti-TNF antibody, isotype control or dexamethasone beginning 2 days prior to infection with 50 arthroconidia of the attenuated *Coccidioides* strain Cp1038. Treatment was continued with dosing every 3–4 days. (**B**) Mice were treated with isotype control or anti-TNF antibody beginning 2 days prior to infection and continuing either 14 days post-infection or throughout the experiment. Data used with permission from Powell et al. [47].

**Figure 5 jof-10-00256-f005:**
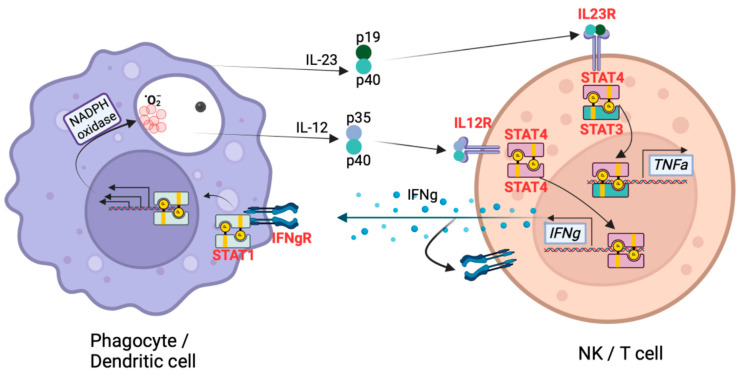
Primary immune deficiencies predisposing to severe coccidioidomycosis center on the innate IL-12/IFNγ pathway with signaling between phagocytic cells and T cells or Natural Killer cells. Pathway components with identified mutations are highlighted in red. Figure created using BioRender.

**Figure 6 jof-10-00256-f006:**
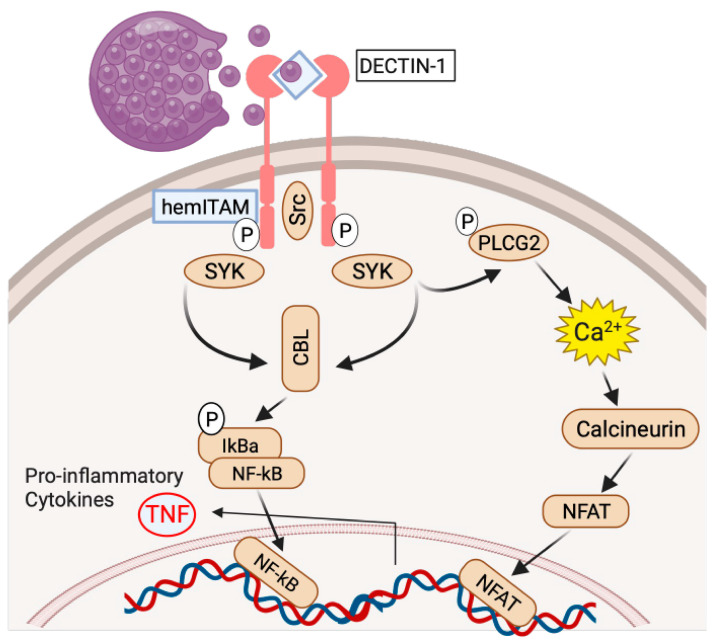
Initial fungal recognition is via the C-type lectin receptor, DECTIN-1. Parallel signaling pathways after β-glucan recognition by DECTIN-1 lead to activation of NFkB and NFAT transcription factors and production of TNF. Figure created using BioRender.

**Figure 7 jof-10-00256-f007:**
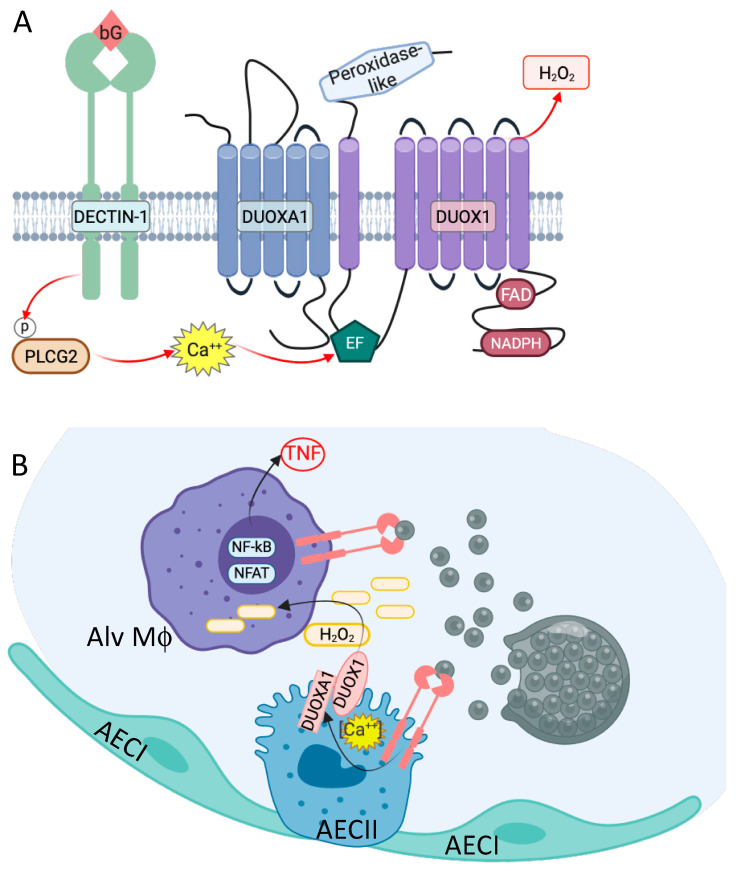
DECTIN-1 signaling induces H_2_O_2_ production by the pulmonary epithelial NADPH oxidase complex, DUOX1/DUOXA1. (**A**) Increased intracellular Ca^++^ after β-glucan recognition by DECTIN-1 is sufficient to activate the intracellular EF-hand domains of DUOX1, leading to release of H_2_O_2_. (**B**) Cartoon of H_2_O_2_-mediated alveolar crosstalk between Type I (AECI) and Type II (AECII) alveolar epithelial cells and alveolar macrophage (Alv Mϕ). Figure created using BioRender.

**Figure 8 jof-10-00256-f008:**
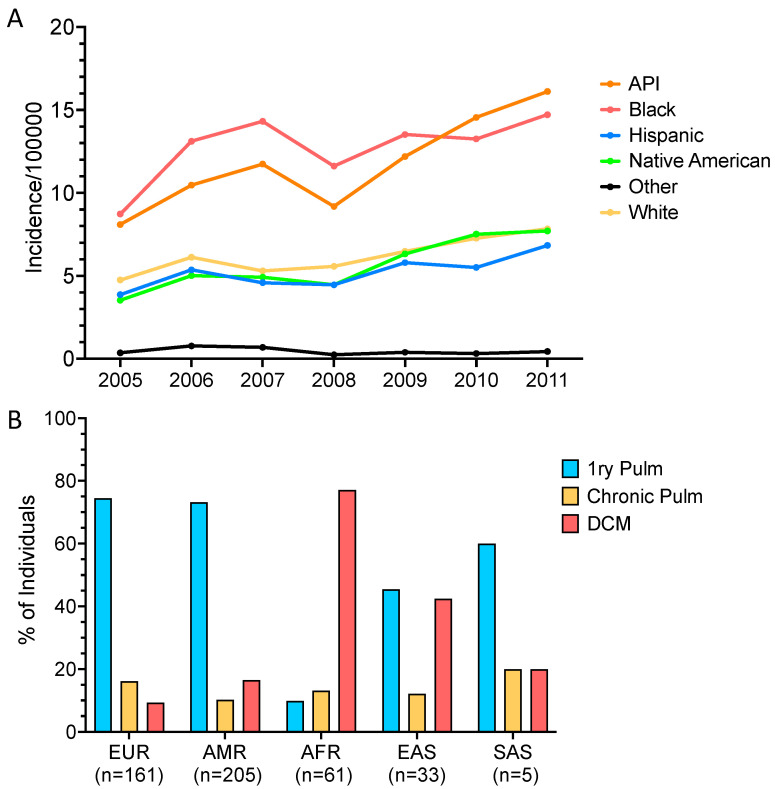
Disease presentation by ancestry. (**A**) Annual Arizona and California coccidioidomycosis hospital incidence rate per 100,000 persons separated by race/ethnicity. Data adapted from Kupferwasser, et al. [92]. API—Asian/Pacific Islander. (**B**) Disease presentation in a California cohort of coccidioidomycosis cases (*n* = 468) separated by genetically determined ancestry. EUR—European; AMR—Admixed American; AFR—African American; EAS—East Asian; SAS—South Asian; 1ry Pulm—Primary pulmonary disease; Chronic pulm—chronic pulmonary disease continuing after >1 yr antifungal therapy; DCM—disseminated coccidioidomycosis.

**Figure 9 jof-10-00256-f009:**
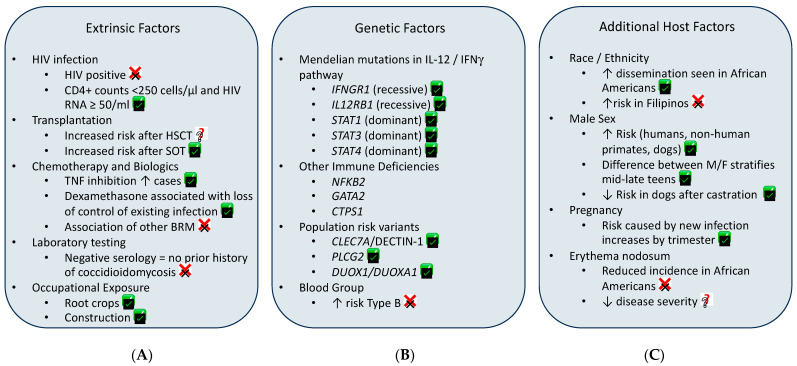
Factors purportedly associated with severe or disseminated coccidioidomycosis discussed in this review. (**A**) Extrinsic medical treatments and occupational exposure. (**B**) Genetic variants reportedly associated with severe disease. (**C**) Additional host factors found in the literature. ✅—Risk factors with strong evidence; ❓—more studies required to validate; ❌—weak evidence to support this as risk factor.

**Table 1 jof-10-00256-t001:** Coccidioidomycosis among HIV-infected individuals.

HIV+ Cases	Active/Positive ^#^	Diffuse Pulmonary Infiltrate (*n*)	DCM	Death	CD4 < 250/µL (*n*)	Reference
27	7	6	5	7	4 (5)	[19]
77	77	31	29 *	32	46 (55)	[11]
170	13	5	1	5	12 (13)	[20]
153	153	80 (146)	43	29	135 ^&^ (153)	[21]
257	29	4 (15)	2	nd	4 (9)	[22]

^#^ Patients developing active coccidioidomycosis or positive laboratory diagnosis; DCM–Documented disseminated disease; * Includes 9 pulmonary infiltrate patients with DCM; Nd—not documented; ^&^ CD4 count < 200/µL.

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
