# Peer review of "The Known and Unknown “Knowns” of Human Susceptibility to Coccidioidomycosis"

_jof, 2024, doi:10.3390/jof10040256_

Round 1
Reviewer 1 Report
Hsu has recently highlighted the known and unknown “knowns” of human susceptibility to Coccidioidomycosis.
Although the author has a catchy title, it may not be clear what unknowns are discussed in the abstract alone. Since it’s a short abstract, the author could expand on this commentary, particularly the disconnect between case reports and scientific research.
The review reads well but could benefit from expanding further. An example is the histology of Figure 1- Could the author further describe the vacuole present in the younger spherule vs. the more developed and segmented spherule that appears to attract neutrophils and phagocytic cells? Is there something unique within the cell wall of mature Cocci that attracts cells? The author can illustrate on the life cycle where each spherule is currently at.
Figure 2 - Could figure 2 mention the duration of illness on the legend with respect to time/weeks/ months?
The author describes NK/ T cells, but the "known" of the protective role of cell-mediated immunity wasn't discussed prior to Figure 5. It would benefit to highlight some recent reviews to illustrate this but could also be addressed. NK cell's role is not well known by non-immunologist this may be beneficial to expand further.
Figure 6 and 7 - I do not believe the coccidioidal ligand is known for Dectin-1 (this could be addressed in the unknown knowns of Coccidioides). Is the amount of b-glucans expressed in inhaled spores, the spherule initials, or endospores known? Can this variability in b-glucan expression be due to SOW masking, thereby Dectin-1 recognition? Additionally, the primary cells that express CLEC7A, according to the Protein Atlas, are primarily macrophages. Can the author discuss the use of databases to understand further point mutations recently identified?
Section 5. Additional Risk factors introduce known risk factors, but the author did not introduce erythema nodosum but did introduce various blood group associations. I would suggest introducing this topic and expanding further. Then, in additional host factor Figure 9, they are not in the order in which they were discussed.
Minor: can you italicize Coccidioides in line 65, discuss the abbreviation of Coccidioides or Cocci on line 72, line 224, line 283
figure 3: the graph quality is lacking, and a bit pixelated may I suggest to change the color of NV,NM,UT and other states? the purple color is very similar
can the author introduce GVHD in line 142?
Figure 4- can the author also mention the dose used for dexamethasone?
double space in line 183, 318
figure 7b not sure this the alveolar macrophage is misslabeled with an F or if it’s meant to be another symbol.
Question for Dr. Hsu: Do you believe there's any role in CARD9 deficiencies in coccidioidomycosis since it is a down-stream adaptor molecule of Dectin-1? Or is the prevalence of Candida sp. infections the reason we associate with CARD9 deficiencies and fungal infections?
Author Response
Reviewer 1.
Is the quality of the figures satisfactory? – No
Since the figure specific comments are repeated below, the responses are addressed there.
Figure 1. Can the author further describe the morphological changes in the parasitic phase? Readers may not be familiar with the unique morphology of the parasitic phase (bottom spherule), which is immature based on the large vascular space shown, while the top spherule shows further segmentation. Perhaps the mature spherule is endosporulating resulting in an influx of phagocytes. This can be highlighted in the legend describing the spherule life cycle.
Figure 2. Can you describe the length of Duration (days/weeks/months). Do we know the average timeline we note severity from CAP?
Figure 3. Timeline is very pixelated and pink and purple color legend are too close to tell apart, I would suggest to change the color choice ( NV, NM, UT vs others)
Figure 4. alpha symbol has changed. Could authors add Day -2 to illustrate earlier timepoint treatment?
Thank you for your comment – as stated in the figure legend, these figures are used with permission from another publication – so changing colors or the alpha symbol is not possible
Figure 5. It's unclear what the role of NK/ T cells are are they were not really introduced in the section.
The point of the figure is to show the IFNg/IL-12 signaling pathway in which causative monogenic mutations have been identified. For early response to Coccidioides, there is not a lot of evidence implicating T-cells. Early response is innate cell mediated. T-cells play a role in developing adaptive and long-lasting immunity as evidenced by RAG knockout mice.
Figure 6. Based of the illustration, it appear that Dectin-1 recognizes endospores. I do not believe the coccidioidal ligand is known to be recognized by Dectin-1 (this could be addressed in the unknown knowns of cocci)
Figure 7. please describe the bG as beta-glucan on figure legend that is recognized by dectin-1
Figure 9. check and cross figures are overlapping with similar symbols, please correct
I’m not certain what is meant by this. In my figure there is no overlap between check and cross figures.
Reviewer 1 specific review points:
Hsu has recently highlighted the known and unknown “knowns” of human susceptibility to Coccidioidomycosis.
Although the author has a catchy title, it may not be clear what unknowns are discussed in the abstract alone. Since it’s a short abstract, the author could expand on this commentary, particularly the disconnect between case reports and scientific research.
The abstract now specifically mentions early, observational studies that have become part of the literature without subsequent rigorous scientific study.
“Early studies were observational, frequently with small numbers of cases; several of these early studies are highly cited in review papers, becoming part of the coccidioidomycosis “canon”. Specific genetic variants, sex, and immune suppression by TNF inhibitors, have been validated in later cohort studies, confirming the original hypotheses. By contrast, some risk factors, such as ABO blood group, Filipino ancestry, or lack of erythema nodosum among black individuals are repeated in the literature despite lack of supporting studies or biologic plausibility.”
The review reads well but could benefit from expanding further. An example is the histology of Figure 1- Could the author further describe the vacuole present in the younger spherule vs. the more developed and segmented spherule that appears to attract neutrophils and phagocytic cells? Is there something unique within the cell wall of mature Cocci that attracts cells? The author can illustrate on the life cycle where each spherule is currently at.
Thank you for this suggestion. The figure has been edited to number the arthroconidia-spherule-endospore-spherule cycle and the stages have been referenced in the edited figure legend.
“Inside the warm, moist environment of the lung the arthroconidia will swell (1), undergo internal cell division (2) develop endospores (3), then rupture (4), releasing endospores which can develop into new spherules. Figure created using BioRender. B. Mouse lung 14 days post infection with C. posadasii . Top, H&E stain showing immune cells swarming the top, rupturing spherule (stage 4 on the left) while lower, unrecognized spherule is less mature (stage 3) as evidenced by the open space inside the spherule.”
Figure 2 - Could figure 2 mention the duration of illness on the legend with respect to time/weeks/ months?
Thank you for this suggestion, the figure has been modified to include rough durations of illness.
The author describes NK/ T cells, but the "known" of the protective role of cell-mediated immunity wasn't discussed prior to Figure 5. It would benefit to highlight some recent reviews to illustrate this but could also be addressed. NK cell's role is not well known by non-immunologist this may be beneficial to expand further.
I appreciate the reviewer’s suggestion. The discussion of T and NK cells was in the context of reported Mendelian mutations in DCM which have centered on the IL-12/IFNg pathway. Prior to this point in the manuscript, there is no discussion of immune cell involvement. The role of NK cells in Coccidioidomycosis is not well characterized. Early studies often cited are based on Leu-11 antibody which recognizes CD16 meaning those studies mentioning NK cells also include monocytes. This issue has been addressed twice – once in the setting of the IL12/IFNg pathway
“Additionally, despite its seeming importance, mutations within this pathway are exceedingly rare. Little is known regarding the role of NK cells during Coccidioides infections.”
And a second time in the section, Unanswered questions,
“On the host side, cells involved in response to the initial infection are not fully characterized. While alveolar macrophages are present and neutrophils arrive rapidly upon spherule rupture, the role of innate lymphoid cells, NK cells and eosinophils have not been characterized. Petkus and Baum [104] report NK cells are the leukocytes responsible for Coccidioides growth inhibition in vitro. This conclusion is hampered by their use of Leu-11 antibody [104] which recognizes CD16 and is found not only on NK cells but on neutrophils, monocytes, and macrophages. “
Figure 6 and 7 - I do not believe the coccidioidal ligand is known for Dectin-1 (this could be addressed in the unknown knowns of Coccidioides). Is the amount of b-glucans expressed in inhaled spores, the spherule initials, or endospores known? Can this variability in b-glucan expression be due to SOW masking, thereby Dectin-1 recognition? Additionally, the primary cells that express CLEC7A, according to the Protein Atlas, are primarily macrophages. Can the author discuss the use of databases to understand further point mutations recently identified?
1,3-B-glucan is a major component of Coccidioides cell walls at all stages of growth as shown by Converse in 1967 and Pappagianis in 1982. DECTIN-1 recognizes both 1,3 and 1,6-B-glucans as shown by Gordon Brown in 2003 (PMID 12719478). Together, this suggests that the cell wall 1,3-B-glucan is the ligand for DECTIN-1. This is further supported by the loss of TNF expression by PBMCs from patients carrying the DECTIN-1 Y238* mutation or the downstream PLCG2 R268W variants after stimulation with purified b-glucan (PMID 36166305).
While macrophages do express DECTIN-1, the primary cells with highest expression are neutrophils (https://www.proteinatlas.org/ENSG00000172243-CLEC7A/immune+cell). That being said, DECTIN-1 is not immune cell specific as shown in the Tissue overview section on Human Protein Atlas (https://www.proteinatlas.org/ENSG00000172243-CLEC7A/tissue). The tissue with the second highest expression is the respiratory system and, within that, alveolar cells represent the highest source of transcript expression – 35-40%. I appreciate the reviewer’s suggestion to discuss the use of databases to understand point mutations, however the body of literature on mutation identification, analysis and demonstration of pathogenicity is huge and is well beyond the scope of this article.
Within the literature, expression of DECTIN-1 on alveolar epithelial cells has been shown by immunohistochemistry (PMID 25161190)
Section 5. Additional Risk factors introduce known risk factors, but the author did not introduce erythema nodosum but did introduce various blood group associations. I would suggest introducing this topic and expanding further. Then, in additional host factor Figure 9, they are not in the order in which they were discussed.
While I appreciate the reviewer’s comment, peer review is designed to address factual errors or errors in methodology/interpretation more so than the reviewer’s desired stylistic format.
Minor: can you italicize Coccidioides in line 65, discuss the abbreviation of Coccidioides or Cocci on line 72, line 224, line 283
figure 3: the graph quality is lacking, and a bit pixelated may I suggest to change the color of NV,NM,UT and other states? the purple color is very similar
The graph color has been changed.
can the author introduce GVHD in line 142?
The abbreviation has been removed
Figure 4- can the author also mention the dose used for dexamethasone?
I refer the reviewer to the publication from which the image was taken and is cited in the figure legend.
double space in line 183, 318
figure 7b not sure this the alveolar macrophage is misslabeled with an F or if it’s meant to be another symbol.
The figure is labeled as Alv MΦ with the Greek symbol for phage and that is defined in the figure legend. I am not certain what “F” the reviewer is referring to - possibly a download issue from the MDPI site.
Question for Dr. Hsu: Do you believe there's any role in CARD9 deficiencies in coccidioidomycosis since it is a down-stream adaptor molecule of Dectin-1? Or is the prevalence of Candida sp. infections the reason we associate with CARD9 deficiencies and fungal infections?
Given the genetics identified in susceptibility to Coccidioidomycosis, it appears the CARD9/BCL-10/MALT1 pathway is less involved than the PLCG2-Ca++ pathway. This may be due to the severity of disease in CARD9 deficiency, most frequently presenting in childhood, (reviewed in https://www.ncbi.nlm.nih.gov/pmc/articles/PMC6157734/), the rarity of CARD9 deficiency, with <100 cases reported worldwide, and finally there is a lack of phenotype in carrier parents and relatives. Together, this does not exclude a role for CARD9 signaling in susceptibility to Cocci, but suggests that the overlap of CARD9 deficiency and Coccidioides infections has not yet happened. It would be surprising if that combination did not result in severe or fatal disseminated disease.
I am not certain why the reviewer believes the manuscript needs extensive editing of English language.
Reviewer 2 Report
I understand the goal of the research work and the direction you have given the manuscript. However, I suggest you take into account expanding a little on what was mentioned above about the basic biology of the fungus, as well as its basic cellular constitution.
In panel A of Figure 1, I suggest you place the name of the certain cellular stages of the microorganism.
In Figure 2. I suggest you place the periods (duration), perhaps weeks or days. Only if possible, because perhaps that depends on the host and his response.
In some parts of the manuscript, you will find the incomplete name of the microorganisms (write the full name of the microorganism if it is the first time it appears in the text) then you can abbreviate it.
Author Response
I understand the goal of the research work and the direction you have given the manuscript. However, I suggest you take into account expanding a little on what was mentioned above about the basic biology of the fungus, as well as its basic cellular constitution.
In panel A of Figure 1, I suggest you place the name of the certain cellular stages of the microorganism.
Thank you for your comment. I have numbered the stages and included them in the figure legend, additionally I note the stages of the spherules in the histology section shown in the panel.
In Figure 2. I suggest you place the periods (duration), perhaps weeks or days. Only if possible, because perhaps that depends on the host and his response.
Thank you for this suggestion. Figure 2 has been edited to include rough duration of symptomatic disease.
In some parts of the manuscript, you will find the incomplete name of the microorganisms (write the full name of the microorganism if it is the first time it appears in the text) then you can abbreviate it.
Thank you for your comments. I have omitted abbreviations and use the full name of the organism or disease throughout.
Reviewer 3 Report
The manuscript “The Known and Unknown "Knowns" of Human Susceptibility to Coccidioidomycosis” reviews the literature about susceptibility to coccidioidomycosis, and the factors that affect this susceptibility. Additionally, it discusses the evidence for commonly reported risk factors. It is an interesting and well-structured work. I also agree with the author that many risk factors influence the acquisition of coccidioidomycosis. However, they are based on old reports with reports of small cases, so it is necessary to carry out studies with more clinical reports and apply solid statistical analysis to identify people at risk of infection with Coccidioides.
Line 65: Change “Coccidiodes” to “Coccidioides”
Line 79: Place at the bottom of Figure 3 the meaning of the state abbreviations
Line 224: Change “Coccidiodes” to “Coccidioides”
Line 283: Change “Coccidiodes” to “Coccidioides”
Line 427: correct the font style “Figure 8. Disease presentation by ancestry. A"
Line 490: Change “Coccidioidal” to “coccidioidal”
Lines 540-543: Correct font style
Lines 591: Correct font style
References
Check references carefully and strictly follow the journal format.
Author Response
The manuscript “The Known and Unknown "Knowns" of Human Susceptibility to Coccidioidomycosis” reviews the literature about susceptibility to coccidioidomycosis, and the factors that affect this susceptibility. Additionally, it discusses the evidence for commonly reported risk factors. It is an interesting and well-structured work. I also agree with the author that many risk factors influence the acquisition of coccidioidomycosis. However, they are based on old reports with reports of small cases, so it is necessary to carry out studies with more clinical reports and apply solid statistical analysis to identify people at risk of infection with Coccidioides.
Line 65: Change “Coccidiodes” to “Coccidioides”
Line 79: Place at the bottom of Figure 3 the meaning of the state abbreviations
Line 224: Change “Coccidiodes” to “Coccidioides”
Line 283: Change “Coccidiodes” to “Coccidioides”
Line 427: correct the font style “Figure 8. Disease presentation by ancestry. A"
Line 490: Change “Coccidioidal” to “coccidioidal”
Lines 540-543: Correct font style
Lines 591: Correct font style
Thank you for your careful reading. The above mistakes have been corrected throughout and the state abbreviations are included in the figure legend of Figure 3. I apologize but I do not understand your “font style” comments for Figure 8 – I looked to see if there is a font change but there isn’t.
References
Check references carefully and strictly follow the journal format.
The journal format has been corrected.
Round 2
Reviewer 1 Report
Dr. Hsu,
Thank you for addressing concerns and improving the figures, the readers will benefit from learning in a succinct manner what are the open questions of CM.
I sincerely apologize for the previous English language and style concern. That was incorrectly selected and was emailed to the editors to please correct it, I am sorry this wasn't relayed to you.
everything has been addressed